# An evidence-based approach to artificial intelligence education for medical students: A systematic review

Nikola Pupic[1], Aryan Ghaffari-zadeh[1*], Ricky Hu[2], Rohit Singla[1], Kathryn Darras[3], Anna Karwowska[4,5], Bruce B. Forster[3]

1 Faculty of Medicine, University of British Columbia, British Columbia, Vancouver, Canada, 2 Faculty of Medicine, Queen's University, Ontario, Kingston, Canada, 3 Faculty of Medicine, Department of Radiology, University of British Columbia, British Columbia, Vancouver, Canada, 4 Association of Faculties of Medicine of Canada, Ontario, Ottawa, Canada, 5 Faculty of Medicine, Department of Pediatrics, University of Ottawa, Ontario, Ottawa, Canada

* sghaff01@student.ubc.ca

**Data Availability Statement:** All relevant data are within the manuscript and its Supporting information files.

## Abstract

The exponential growth of artificial intelligence (AI) in the last two decades has been recognized by many as an opportunity to improve the quality of patient care. However, medical education systems have been slow to adapt to the age of AI, resulting in a paucity of AI-specific education in medical schools. The purpose of this systematic review is to evaluate the current evidence-based recommendations for the inclusion of an AI education curriculum in undergraduate medicine. Six databases were searched from inception to April 23, 2022 for cross sectional and cohort studies of fair quality or higher on the Newcastle-Ottawa scale, systematic, scoping, and integrative reviews, randomized controlled trials, and Delphi studies about AI education in undergraduate medical programs. The search yielded 991 results, of which 27 met all the criteria and seven more were included using reference mining. Despite the limitations of a high degree of heterogeneity among the study types and a lack of follow-up studies evaluating the impacts of current AI strategies, a thematic analysis of the key AI principles identified six themes needed for a successful implementation of AI in medical school curricula. These themes include ethics, theory and application, communication, collaboration, quality improvement, and perception and attitude. The themes of ethics, theory and application, and communication were further divided into subthemes, including patient-centric and data-centric ethics; knowledge for practice and knowledge for communication; and communication for clinical decision-making, communication for implementation, and communication for knowledge dissemination. Based on the survey studies, medical professionals and students, who generally have a low baseline knowledge of AI, have been strong supporters of adding formal AI education into medical curricula, suggesting more research needs to be done to push this agenda forward.

**Funding:** The authors received no specific funding for this work.

**Competing interests:** I have read the journal's policy and the authors of this manuscript have the following competing interests: BF is a Senior Deputy Editor of Journal of Canadian Association of Radiologists, and Associate Editor of British Journal of Sports Medicine. All other authors declare no competing interests related to this manuscript.

## Author summary

Artificial intelligence (AI) has grown exponentially in the last two decades, presenting itself as a significant opportunity to improve the quality of patient care. Despite these advancements, medical programs have been slow to adapt and incorporate AI, which has resulted in a lack of AI-specific teaching in medical schools. Our work evaluates the reasons for why AI should be included in medical school programs to prepare learners for a future where they will have to interact with AI. Based on our analysis, we identified six themes that would be crucial to teach in order to successfully implement AI in medical schools and graduate AI-competent physicians, including: ethics, theory and application, communication, collaboration, quality improvement, and perception and attitude. The general atmosphere in the medical community agrees with our stance, with strong support among medical professionals and students for the inclusion of formal AI education in medical curricula. This highlights the importance of this topic and the need for further research to advance our AI agenda.

## 1. Introduction

Artificial intelligence (AI) applications in medicine have undergone exponential growth in the past two decades [1]. AI, a general term, implies "the use of a computer to model intelligent behaviour with minimal human intervention" [2]. For instance, in radiology, computer-aided tools can help in the detection of pneumonia or assist in liver and tumor segmentation. These programs aim to assist physicians in the detection and classification of disease, resulting in higher accuracy, reduced variability, or faster results. Despite its rapid emergence, medical training programs have yet to adapt and include AI education as a standard core component [3]. It is now reasonable to assume that graduating medical students should have a basic understanding of AI and how it can improve patient care.

Numerous experts have advocated for the incorporation of AI training and literacy into medical curricula [3–6]. Paranjape *et al.* and Wartman and Combs believe that clinical practice is changing from the information age to the AI age [3,4]. This shift promotes abstraction, which would require physicians to employ AI to manage the wealth of information without memorising it, allowing them to prioritize important tasks. However, abstraction requires education to contextualize AI's capabilities. McCoy et al. state that graduating physicians do not need to understand the complexities of AI algorithms, but they must know when they are applicable and the strengths and weaknesses of the data output.[6] These authors also differentiate between the knowledge that all physicians need for everyday practice versus what some physicians need to drive innovation, highlighting that a curricular reform should address the former and extracurricular programs the latter [6]. Support for a curricular reform has been echoed through the medical world, with Banerjee et al. and Teng et al., reporting that 81% of physicians and 63·36% of medical students, respectively, support fundamental AI literacy training [7,8]. Scheetz et al. reported that although physicians are aware of AI as a concept, only 5·5% report their knowledge as excellent, further suggesting the need for a curricular reform [9].

Several groups have investigated implementing ad-hoc extracurricular programs to improve AI literacy, such as the workshops led by Hu et al. and the Artificial Intelligence Curriculum for Residents (AI-RADS) program at Dartmouth [10,11]. These programs, and others, are mostly extracurricular and vary significantly in what is included in their proposed curricula.

The implementation of extracurricular training programs signals that AI training for physicians is at an early stage. Due to issues like topic selection, there is no unified medical curriculum. Despite the support for including AI education, financial barriers, resistance of educators to change curricula, lack of staff with teaching experience in AI, and limited digitalization are some of the main barriers for implementing AI into the medical curriculum [3,10,12,13]. The purpose of this systematic review is to identify and aggregate the current evidence-based recommendations for the development and implementation of an AI curriculum in undergraduate medical education (UGME). We hope that exposing medical students to an AI curriculum in the future will result in an increased integration of AI into their future practice and improved competency and confidence when using AI in medicine.

## 2. Methods overview

This systematic review followed PRISMA (Preferred Reporting Items for Systematic Review and Meta-analysis) guidelines [14]. This systematic review also used the Newcastle-Ottawa scale (NOS) and the Risk of Bias in Systematic Reviews (ROBIS) tools to assess bias [15,16].

### Search strategy

MEDLINE, EMBASE, CINAHL, ERIC, NCBI, and Web of Science were searched from database inception to April 23, 2022, for articles addressing AI education in undergraduate medical education. To guide our search, the search terms of "medical education", "artificial intelligence", "medical curriculum", and "medical program" were used combined with Boolean operators "AND", "OR", and "ADJACENT".

### Assessment of study eligibility

The inclusion and exclusion criteria for this study were determined a priori. Criteria for inclusion were: studies about undergraduate medical students, cross sectional and cohort studies assessed as fair quality or higher by NOS, systematic and scoping reviews, randomized controlled trials (RCTs), and Delphi studies.[15] Based on the NOS, fair quality evidence includes 2 stars in the selection domain AND 1 or 2 stars in the comparability domain AND 2 or 3 stars in the outcome/exposure domain [15]. Exclusion criteria were: non-English studies, post-graduate medical education studies, papers targeting allied healthcare workers, original research that focuses on applications of AI and not AI education, conference abstracts, case reports, narrative studies, studies assessed as poor quality using the NOS. Based on the NOS, poor quality evidence includes 0 or 1 star in the selection domain OR 0 stars in the comparability domain OR 0 or 1 stars in the outcome/exposure domain [15].

### Study screening

Abstract and full-text screening were performed independently and in duplicate by two reviewers (NP and AG) using the Covidence platform (Veritas Health Innovation Ltd., Melbourne, Australia). Any conflicts that arose when consensus was not reached during the screening steps were resolved by a third reviewer (RS).

### Data extraction

Data extraction was performed in Excel (Microsoft, Washington, USA). The studies were subdivided into two categories of: survey-based studies and non-survey studies.

For all studies, the following data fields were extracted manually: title, author, country, year of publication, study type, population, sample size, AI principles focused on, any previous

exposure to AI, years of experience working with AI, male to female ratio, education level of respondents, factors that may influence respondents' answers (i.e., anonymized responses, closed-ended questions, open-ended questions), and the study's level of evidence. Specific to survey studies, additional information extracted included: survey validation status, number of respondents, types of questions asked, survey structure, outcomes evaluated, AI principles focused on, AI exposure of respondents, framework for questions (structured or unstructured), sampling method, and prior exposure to AI. Finally, for non-survey studies, the additional fields extracted were type of validation, sampling type, sensitivity, specificity, key results reported, outcomes evaluated, effect size measurements, statistical significance, clinical significance, and type of intervention.

Data extraction was checked for accuracy and completion by a third reviewer (RS).

### Additional references

The reference lists of all included articles were subsequently reviewed for additional relevant articles. Eight additional articles were included.

### Thematic analysis

Using the extracted data, two reviewers (NP and AG) created a MindMap using Mural (Mind Mapping Software, Buenos Aires, Argentina) to group various themes that were identified throughout the extraction process. Each group contained information supporting its status using evidence from the extracted data.

### Quality assessment and statistical analysis

Quality assessment was performed by two reviewers (NP and AG) using the NOS criteria for cross sectional and cohort studies and the ROBIS criteria for reviews [15,16]. The NOS evaluates studies based on the selection of the study groups, the comparability of groups, and the ascertainment of either the exposure or outcome of interest [15]. A meta-analysis of the included articles was not performed due to the significant heterogeneity in the study types included. However, the article data was categorized based on the frequency of keywords related to the themes identified in the thematic analysis.

## 3. Results

The original search yielded 991 studies after duplicates were removed. A total of 34 studies were included for analysis after title, abstract, full-text screen, and reference mining (Fig 1). Of the 34 total included studies, 23 were evaluated with the NOS criteria and 11 with the ROBIS criteria. One study met the NOS criteria for good [17] and 22 studies met the NOS criteria for fair [7,8,18–36,37]. Seven studies met the ROBIS criteria for low risk [12,13,38,39,40,41,42], one study was of unclear risk [43], and three studies were of high risk [44–46]. No RCTs were published within our search criteria. Table 1 provides a summary of the studies included. S1 and S2 Tables in the supplemental documents show an overview of the NOS scoring for cross-sectional and cohort studies and ROBIS scoring for reviews, respectively. The studies were separated into two categories: survey (n = 23) and non-survey (n = 12). The mean sample size of the survey studies was 578 (range 62–1459). The mean respondent years of experience working with AI was only reported in three studies [7,21,28]. The mean male to female ratio of respondents was 54% female and 46% male. A thematic analysis of the included studies revealed six themes: ethics, theory and application, communication, teamwork, quality improvement, and

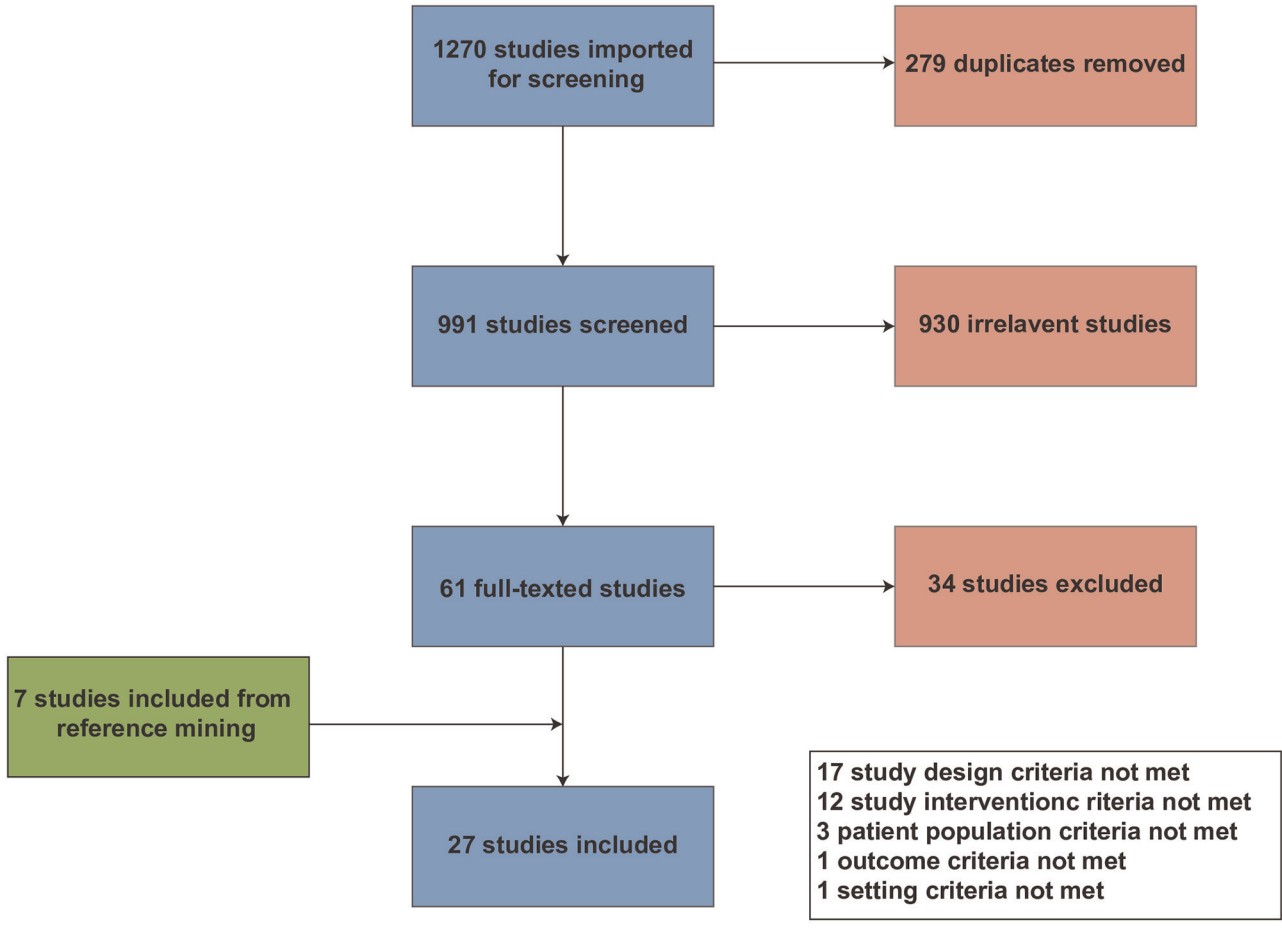

**Fig 1. PRISMA diagram depicting the screening process undertaken.**

perception and attitude. Fig 2 lists the percentage of total studies supporting each theme. This section also discusses each topic and applicable sub-themes with supporting evidence.

Medical AI ethics divides into patient-centric and data-centric ethics. Patient-centric ethics refers to the safe and effective implementation of AI to build awareness of patient inclusion, rights, and equity while addressing bias [13,45]. For example, ensuring that data collection and algorithm design are made equitable by collecting representative training sets to minimize bias [13,37]. Data-centric ethics refers to the legal aspects of safe data handling to preserve patient privacy and to prevent data from being compromised [13]. It requires algorithmic validation, using techniques such as federated learning or subgroup analysis, to ensure they are controlled for bias [37,45], and that the intellectual property over the algorithms is protected by copyright [13,46].

Two sub-themes were identified within theory and application: knowledge for practice and knowledge for development. Knowledge for practice refers to first establishing a strong foundation of basic statistics that would help facilitate learning the basic AI specific knowledge needed to effectively use AI tools in a clinical environment [12,13,17,36,39,42,44,45]. Overall, this includes learning the AI terminology [13,37,45], strengths and limitations of AI [12,13,16,37,42], risks of AI [12,13,17,37,39,42], controlling for bias [42], critical appraisal AI literature and tools [12,13,34,37,39,45,46], and shared decision-making [12,13,37,41,45]. In

**Table 1. A summary of the included studies in the review.** The studies evaluated the current state of AI education in undergraduate medical curricula, as well as the perception, attitude, and knowledge of undergraduate medical students and physicians regarding AI, the current state of AI education in medicine, and the impact of AI on the future of medicine. ML = machine learning, DL = deep learning, EHR = electronic health records.

| Author(s) | Study Type | Year Published | Country | Sample Size | Male to Female Ratio | Education Level | Outcomes Evaluated/ Questions Asked | Key Results Reported | NOS Score | ROBIS Score |
|---|---|---|---|---|---|---|---|---|---|---|
| Banerjee et al. [7] | Survey Study | 2021 | UK | 210 | 47% female | Trainee doctors | Training in AI and interaction with AI systems | 92% reported insufficient training in AI in their current training, 81% supported more formal training in AI, 62% believed AI would reduce clinical workload | 5 | N/a |
| Teng et al. [8] | Survey Study | 2022 | Canada | 2167 | 62·53% female | Medical students and physicians | Demographics, understanding of AI, attitude towards the impact of AI, priorities in AI literacy and education | 63·36% believed that gaining basic literacy in AI should be part of their curriculum, 29·44% preferred either a multiple-workshop series, 16·11% preferred a one-day course. | 5 | N/a |
| Lee et al. [12] | Scoping review | 2021 | Canada | N/A | N/A | N/A | Identify key themes and gaps on how to train and prepare students for using AI in clinical practice | Curriculum delivery should include experiential learning, could also include modules and small group sessions. There are barriers to implementation such as faculty resistance, lack of AI accreditation and licensing, lack of AI core competencies, lack of faculty expertise on AI, and lack of awareness regarding how AI will impact the future of healthcare. | N/a | Low Risk |

*(Continued)*

**Table 1.** (Continued)

| Author(s) | Study Type | Year Published | Country | Sample Size | Male to Female Ratio | Education Level | Outcomes Evaluated/ Questions Asked | Key Results Reported | NOS Score | ROBIS Score |
|---|---|---|---|---|---|---|---|---|---|---|
| Charow et al. [13] | Scoping Review | 2021 | Canada | N/A | N/A | N/A | Assess current and past AI education programs to inform curricular content, delivery, and effectiveness | There are 3 main barriers to AI adaption: regulatory, economic, and organizational culture. 10 studies present 13 unique programs teaching AI and many are not specifically designed for undergraduate medical education. The cognitive aspect of AI education includes things needed to know, psychomotor involves skills needed to adapt and master, and the affective component considers that attitude needed to develop and incorporate into practice. | N/a | Low Risk |
| Karaca et al. [17] | Intervention Study | 2021 | Turkey | 897 | N/A | Undergraduate medical students | Cognition, ability, ethics, vision— explain 50·9% of the cumulative variance | The Medical Artificial Intelligence Readiness Scale for Medical Students (MAIRS-MS) could be used as an effective screening tool for evaluation and monitoring of medical students' readiness on the topic of AI. | 6 | N/a |
| Gong et al. [18] | Survey Study | 2018 | Canada | 322 | N/A | Medical students | Impact of AI in radiology in terms of training, workload, and employment | 29·3% of respondents agreed AI would replace radiologists in foreseeable future, 67·7% agreed AI would reduce the demand for radiologists, 48·6% agreed AI caused anxiety when considering the radiology specialty | 5 | N/a |
| dos Santos et al. [19] | Survey Study | 2019 | Germany | 263 | 63·8% female | Medical students | Awareness of AI in the context of radiology, current uses of AI in medicine | 52% were aware of AI in radiology, 68% were unaware of the AI technologies involved, 71% agreed on the need for AI to be included in medical training | 4 | N/a |

(*Continued*)

**Table 1.** (Continued)

| Author(s) | Study Type | Year Published | Country | Sample Size | Male to Female Ratio | Education Level | Outcomes Evaluated/ Questions Asked | Key Results Reported | NOS Score | ROBIS Score |
|---|---|---|---|---|---|---|---|---|---|---|
| Sit et al. [20] | Survey Study | 2020 | UK | 484 | N/A | Medical students | AI role in in healthcare future, AI impacting specialty choice, current understanding in AI comfort with nomenclature associated with AI, benefits of AI teaching | 89% believed that teaching in AI would be beneficial for their careers, 78% agreed that students should receive training in AI as part of their medical degree | 6 | N/a |
| Park et al. [21] | Survey Study | 2020 | USA | 156 | N/A | Medical students | Medical students' perception towards AI, primary source of information regarding AI in medicine, AI impact on their enthusiasm for choosing a specialty | 75% agreed that AI would have a significant role in the future of medicine, 66% agreed that diagnostic radiology would be the specialty most greatly affected, 44% reported that AI made them less enthusiastic about radiology | 5 | N/a |
| Reeder and Lee [22] | Survey Study | 2021 | USA | 463 | 43·2% female | Medical students | Impact of AI on radiology ranking, Opinions on radiology and AI, Exposure to radiology and AI, Methods for AI education | 40% of students expressed a concern towards choosing radiology due to AI, 51% of students predicted a decrease in radiology job opportunities due to AI | 6 | N/a |
| Alelyani et al. [23] | Survey Study | 2021 | Saudi Arabia | 714 | 54·6% female | Radiologists, technicians, and radiological science students | AI awareness, AI practices, AI validation, AI outcomes | 81·9% believe that Artificial intelligence must be included in the curriculum and training of medicine and health sciences colleges. | 5 | N/a |
| Auloge et al. [24] | Survey Study | 2020 | France | 1459 | 65% female | Medical students | The future of radiology, awareness regarding interventional radiology, the implementation of AI in radiology. | 65% believe AI is not a threat to radiology | 5 | N/a |
| Doganer et al. [25] | Cross-Sectional Survey Study | 2021 | Turkey | 550 | 66·2% female | Undergraduate students in health sciences (not limited to Medicine) | Use of AI, effects of AI in the future of medicine, effects of AI on business life in the future | Medical students think that artificial intelligence will increase unemployment and will have a negative sociological impact. | 4 | N/a |

(*Continued*)

**Table 1.** (Continued)

| Author(s) | Study Type | Year Published | Country | Sample Size | Male to Female Ratio | Education Level | Outcomes Evaluated/ Questions Asked | Key Results Reported | NOS Score | ROBIS Score |
|---|---|---|---|---|---|---|---|---|---|---|
| Ejaz et al. [26] | Survey Study and Focus Group | 2022 | UK | 128 | 56% female | Medical students | AI knowledge, approach to learning AI, applications of AI in clinical medicine, patient safety | 86% were interested in exploring interdisciplinary learning with engineering, 92% expressed that AI-related teaching needs to be incorporated into the core medical curriculum | 5 | N/a |
| Gillissen et al. [27] | Survey and Discussion Groups | 2022 | Switzerland | 1053 | 74% female | Medical students | Digitization of patient information, digitization of doctor-patient interaction, demographics, and learning | Students within a Case Based Learning (CBL) curriculum believe that AI solutions result in better diagnosis | 5 | N/a |
| Blease et al. [28] | Cross-Sectional Survey Study | 2022 | Ireland | 252 | 62·6% female | Medical students | Familiarity with AI, exposure to AI education | 62·4% of students stated there was 0 hours of training in AI. 48·8% of students somewhat agreed, 18·6% moderately agreed, and 11·6% strongly agreed that AI should be a part of their medical program training. | 4 | N/a |
| van Hoek et al. [29] | Survey Study | 2019 | Switzerland | 170 | 40% female | Doctors, surgeons, and students | AI exposure, how AI should be learned, opinions on AI and radiology | AI significantly lowered students' preference for ranking radiology and it was significantly associated with a lower understanding of radiology. Curricular integration was the preferred method of students for teaching AI. | 6 | N/a |
| Wood et al. [30] | Survey Study | 2021 | USA | 161 | 33% female | Medical students and clinical faculty | Participant's background, AI awareness, and AI applications in medicine. | Students were more interested in AI in patient care training (28% vs 14%), while faculty were more interested in AI in teaching training (16% vs 2%). | 5 | N/a |

(*Continued*)

**Table 1.** (Continued)

| Author(s) | Study Type | Year Published | Country | Sample Size | Male to Female Ratio | Education Level | Outcomes Evaluated/ Questions Asked | Key Results Reported | NOS Score | ROBIS Score |
|---|---|---|---|---|---|---|---|---|---|---|
| Blease et al. [31] | Survey Study and Open Commentary | 2019 | UK | 66 | 42·4% female | General practitioners | Participants were asked to provide any comments on the survey topics seen in study 29. | Eight themes were identified, including empathy and communication, clinical reasoning, patient-centredness, improved efficiency, administrative roles, understaffing, acceptability of AI, and ethics of innovation. | 5 | N/a |
| Blease et al. [32] | Cross-Sectional Survey Study | 2018 | UK | 720 | 44·9% female | General practitioners | Could AI replace general practitioners in analyzing patient data, reaching a diagnosis, and providing empathetic care | The majority of GPs thought it was unlikely for technology to replace physicians in diagnosing patients (68%), creating personalized treatment plans (61%), and providing empathetic care (94%). | 5 | N/a |
| Blease et al. [33] | Delphi Method Study | 2020 | USA | 29 | Round 1: 25% female Round 2 and 3: 31% female | Leading health Informaticians | Forecasts of the impact of AI/ML on patient care, access to care, and the long-future for primary care physicians | Experts anticipated that by 2029 workplace changes within healthcare would require increased AI/ML training for medical students. | 4 | N/a |
| Blacketer et al. [34] | Cross-Sectional Study and Survey Study | 2021 | Australia | 245 | N/A | Undergraduate medical students | ML Knowledge | Students performed poorly on questions related to study design and knowledge questions. However, they performed well on interpreting conclusions and statistical significance of ML research. | 5 | N/a |
| Kansal et al. [35] | Cross-Sectional Study | 2019 | India | 212 | 40·6% female | Medical students and doctors | Association between knowledge of AI, gender, medical experience, etc. as well as knowledge and interest in AI | 74·4% of participants were not knowledgeable about AI and medical students were more interested in learning about AI than physicians (69·3% vs. 51·6%). Female students were significantly less knowledgeable in AI than male students but they were significantly more interested in learning about AI in medicine. | 6 | N/a |

(*Continued*)

**Table 1.** (Continued)

| Author(s) | Study Type | Year Published | Country | Sample Size | Male to Female Ratio | Education Level | Outcomes Evaluated/ Questions Asked | Key Results Reported | NOS Score | ROBIS Score |
|---|---|---|---|---|---|---|---|---|---|---|
| Giunti et al. [36] | Cross-Sectional Descriptive Study | 2019 | Finland | N/A | N/A | Medical students | N/A | 30% of schools within 28 members of the European Union offered a type of course within information technology and 64·4% of them made the course mandatory. | 5 | N/a |
| Harshana Liyanage et al. [37] | Delphi Method Study | 2019 | Germany | 20 experts | N/A | Panelists with previous exposure to AI | Identify stakeholders' perceptions, issues, and challenges surrounding AI in primary care | Primary care community needs to be proactive in guiding the ethical development of AI applications. There should be a formal process to develop an ethics committee that could assess the ethical processing of data in AI applications | 5 | N/a |
| Yang et al. [38] | Scoping Review | 2022 | Canada | N/A | N/A | N/A | Examines stakeholders' perspectives on the use of AI in radiology | Seven themes of AI in radiology were identified, including predicted impact of AI on radiology, potential replacement of radiologists, trust in AI, knowledge of AI, education on AI, economic considerations, and medicolegal implications. | N/a | Low Risk |
| Sapci and Sapci [39] | Integrative Review | 2020 | USA | N/A | N/A | N/A | Examine and evaluate the current state of AI training in medicine | 10/26 papers evaluated AI education in medicine. AI in healthcare is still an emerging field that does not have much high quality evidence. Most studies were either case reports or opinion pieces | N/a | Low Risk |
| Eui-Ryoung Han et al. [40] | Integrative Review | 2019 | Korea | N/A | N/A | N/A | Identify and evaluate the themes that need to be implemented in AI curricula for medical education | Mainly focuses on medical education trends fostering a connection with advanced technologies amongst physicians considering integrative themes in education such as humanistic approach to patient safety, early experience and longitudinal integration, and student-driven learning with advanced technology | N/a | Low Risk |

(*Continued*)

**Table 1.** (Continued)

| Author(s) | Study Type | Year Published | Country | Sample Size | Male to Female Ratio | Education Level | Outcomes Evaluated/ Questions Asked | Key Results Reported | NOS Score | ROBIS Score |
|---|---|---|---|---|---|---|---|---|---|---|
| Lorainne Tudor Car et al. [41] | Review | 2021 | Singapore | N/A | N/A | N/A | Analyze the current digital health education for medical students and identify curricular changes that need to be improved | Does not focus on AI and what encompasses digital health is discussed. Courses focusing on digital health within undergraduate medical curriculum were heterogeneous in content and teaching time. | N/a | Low Risk |
| Grunhut et al. [42] | Integrative review | 2021 | USA | N/A | N/A | N/A | Evaluate the attitudes of medical students towards AI and how to approach implementing AI into the medical curriculum | In UME programs, there should be a focus on medical students developing the skill to create validated information for AI systems and learn about the capabilities of AI. There is an overall lack of implementation of AI within medical education even though there is a general consensus as to a need for AI principles being a part of medical curriculum. | N/a | Low Risk |
| Chan and Zary [43] | Integrative Review | 2019 | UAE | N/A | N/A | N/A | Review current use of AI in medical education, identify challenges of implementing AI in medical education | The primary use of AI in medical education was for learning support. To better integrate Ai into the medical profession, AI should be integrated into the medical school curriculum. | N/a | Unclear Risk |
| Maksut Senbekov et al. [44] | Review | 2020 | Kazakhstan | N/A | N/A | N/A | Discuss and analyze recent changes in digital health topics including AI and medical education | AI can be used for clinical decision making and care management as well as proactive detection to forecast hospital mortality. There has not been much effort in the past for revising current curricula. AI-based training in medical education can complement and enrich the curriculum so students know how to apply AI tools to clinical problems. | N/a | High Risk |

(*Continued*)

**Table 1.** (Continued)

| Author(s) | Study Type | Year Published | Country | Sample Size | Male to Female Ratio | Education Level | Outcomes Evaluated/ Questions Asked | Key Results Reported | NOS Score | ROBIS Score |
|---|---|---|---|---|---|---|---|---|---|---|
| Mark P. Khurana et al. [45] | Delphi Method Study and Scoping Review | 2022 | Denmark | 18 experts | N/A | Digital health experts | Knowledge, skills, and attitude | Attitude towards digital health and basic understanding of digital health are significantly more important than practical skills within digital health. | N/a | High Risk |
| Santomartino et al. [46] | Systematic Review | 2022 | USA | N/A | N/A | N/A | Evaluate and summarizes the attitudes of medical stakeholders' toward the role and impact of AI on radiology | Medical students and radiologists favored the inclusion of implementing AI solutions in medicine with an overall optimism about the integration of AI within radiology. | N/a | High Risk |

terms of implementation, developing the ability to visualize how the role and workflow of a physician may change with AI during UGME [13,33], and learning implementation techniques that would help with the AI transition, may improve the future integration of AI in healthcare.

Knowledge for development refers to what physicians should know to better contextualize AI in the clinical environment, communicate to data scientists and allied healthcare

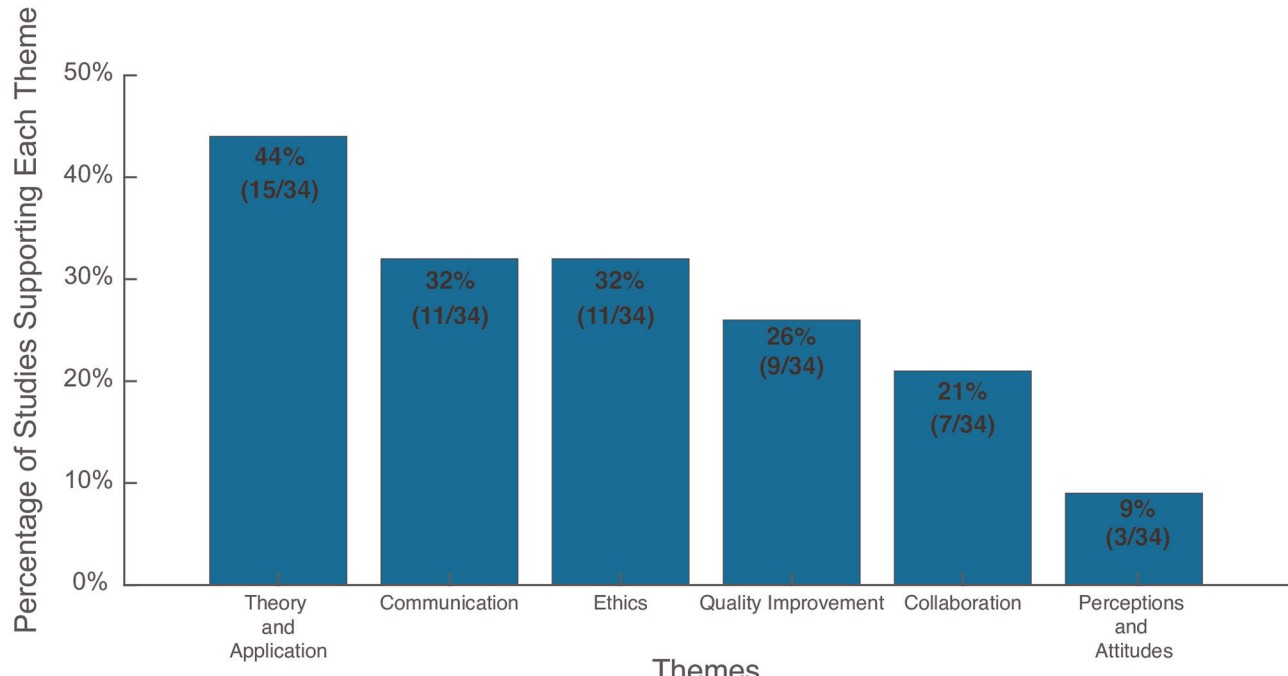

**Fig 2. Bar graph depicting the percentage of total studies that support each of the six themes.**

professionals, and actively contribute to the evaluation, design, and integration of new AI tools [12,13,44]. Eleven studies suggest that physicians should understand the role of data science in AI [12,13,16,37,39,41,42,45]. In particular, knowledge related to data stewardship [12,13,37,41], data preprocessing and acquisition [13], standardization [13], data analytics [13,17,39], health data infrastructure [45], programming [39,41], and big data [13,45], may help physicians appreciate the obstacles faced by data scientists and improve the multidisciplinary collaboration [13,39]. Other important skills include knowledge related to machine learning (ML) [12,13], deep learning (DL) [13], natural language processing (NLP) [13], model development [13], and AI tool design and development [13,39].

Three key types of communication were identified: communication for clinical decision-making [13], communication for implementation [13,37], and communication for knowledge dissemination [12,13,37,41,42]. Furthermore, four key relationships for communication were identified: data scientist to physician [13], physician to physician/allied healthcare professional [12,13,42], physician to patient [12,13,37,41,42], and physician to payer [13,37].

Communication for clinical decision-making focuses on ameliorating a physician's own ability to understand how AI tools are designed, including open communication about the data collection, preprocessing, and acquisition [13], as well as what factors the AI tool uses to make decisions [13]. The goal of this communication is to equip physicians with the tools needed to successfully use AI in a clinical setting [12,13,16,40,44,45]. The key relationships for communication for practice are data scientist to physician and physician to physician/allied healthcare professional.

Communication for implementation focuses on understanding how physician roles and workflows will change when AI gets implemented [13,33]. It also focuses on the economic effects of implementing AI tools and how to communicate with stakeholders [13,37]. The key relationships involved will be physician to physician/allied healthcare professional and physician to payer.

Communication for knowledge dissemination focuses on sharing knowledge to improve the understanding of AI tools with respect to their role in care, as well as the strengths, limitations, and risks of AI [12,13,17,37,39,42]. It also includes the need to show empathy and patience when communicating with and educating others about AI [13,33,40]. The key relationships involved will be physician to physician/allied healthcare professional and physician to patient.

The implementation of AI in medicine requires a multidisciplinary approach, integrating perspectives from developers and clinicians [13]. However, regulatory, economic, and organizational fragmentation between clinical and data science domains inhibits the adoption of AI [13]. Regulatory challenges include integrating AI into clinical settings and workflow, data sharing between institutions, and the translation and interpretation of AI models in a health- care setting [3,13,43]. From an economic perspective, learning about the various economic considerations with respect to AI is crucial to having an understanding of the business and clinical aspects of AI implementation [13]. From an organizational culture perspective, including legal strategies and governance in programs can build awareness regarding the ethics, inclusion, equity, patent rights, and confidentiality when using AI tools [3,5,13].

Quality improvement was reflected through the need for critical appraisal of AI literature and tools [12,13,34,37,39], physician involvement in AI tool design/development [13,39], the ability to identify and control for bias [45], communication with data scientists and engineers [13], and communication with patients [12,13,37,41,42]. Quality improvement was also reflected by the need for evaluating and iterating on any AI curriculum designed for medical schools on a regular basis to further improve the learning experience [12].

The perceptions and attitudes of medical students and professionals reflected some hesitancy surrounding the impacts of AI on medicine, such as unemployment [25], but ultimately showed overwhelming support in favour of AI [7–9,26]. For example, Ejaz et al. report that despite 43% of participants expressing worry regarding the impact of AI on medicine, 92% expressed that AI-related teaching needs to be incorporated into the core medical curriculum [26]. The same authors also report that 86% of students were interested in more interdisciplinary learning within the computer science and biomedical engineering fields [26].

## 4. Discussion

This is the first study that examines the evidence-based recommendations for implementation of AI educational pro- grams in the global undergraduate medical curriculum. Through systematically evaluating various aspects of AI, we were able to look at the current trends in multiple domains of AI curricula including ethical, technical, and economic points of view.

Nine studies discuss the need for a fundamental understanding of AI ethics to improve the adoption of AI in medicine and ensure its safe use [12,13,16,37,39,45]. Charow et al. support the need for AI-specific ethics to be taught in medical schools to address ethical, legal, and data governance issues [13]. Charow et al. also suggest that this knowledge gap can be bridged by establishing a regulatory body that is tasked with creating systematic guidelines for AI ethics [13]. This regulatory body would also monitor data processing, regulate data sharing between institutions, and oversee the general implementation of AI in healthcare [13]. Lee et al. support this sentiment, suggesting the need for the development of a standardized list of core AI competencies and the implementation of an AI curriculum to medical schools [12]. With a regulatory body and a standardized curriculum, medical schools could ensure that future practitioners are able to recognize and identify ethical and legal issues that may arise such as patient confidentiality [13], equity [13,45], inclusion [13], and patient rights [13]. Furthermore, by providing medical students with an opportunity to learn about AI ethics before they enter their clinical experiences, it could not only better prepare them to work with AI, but also better communicate to patients about AI [12,40].

We also found that establishing a strong theoretical base of AI for practice, including its strengths and limitations, is what we should strive for when educating future learners. This would equip them with the skills needed to critically appraise AI literature and tools before implementing them [12,13,34,37,39], interpret model output [13,37,42], and communicate AI findings to others [12,13,42]. Due to the inability for contextualization of AI algorithms, physicians will still need to regularly conduct contextual analyses when handling the care of patients to supplement the AI. Although important, knowledge for development is likely more suited for select individuals who are more passionate for AI and would like to play a larger role in its advocacy and implementation.

Unfortunately, studies did not always clarify specific issues within each area. Many studies advocated for fundamental statistics knowledge but did not define what is entailed. AI is advanced predictive statistics, therefore medical students could consider it part of their biostatistics coursework. Studies suggest that ML, DL, and NLP should be included in the early years of training in undergraduate medical education in order to increase student exposure to AI [12,13]. Post-graduate medical education can then develop these talents further. Communication for clinical decision-making and communication for knowledge dissemination are both patient-facing skills that would be used on a daily basis in clinics. Therefore, focusing on these skills to bridge the gap between a learner's knowledge of the theory and application of AI and providing care should be a primary objective taught to medical students. This would allow physicians to explain the strengths and limitations of AI in a way that patients can understand.

AI-interested physicians can also focus on implementation communication, using the skills they developed during UGME and elsewhere to better connect AI with healthcare. Predicting workflow changes, economic implications, and stakeholder communication would be more logistical. Interdisciplinary engagement with data scientists would improve AI infrastructure by solving regulatory, economic, and organisational obstacles and applying quality improvement techniques for constant monitoring. This role's knowledge may vary by country and healthcare system (i.e., public vs private payer healthcare system). Although there is hesitancy from students surrounding AI implementation, most can see the benefits outweighing the risks, leading to a positive attitude toward learning about AI. As medical school curricula are already saturated with content [47], it may be difficult to move forward without gaining the support of these stakeholders. In addition, there needs to be capacity-building to increase the pool of qualified instructors as those already teaching is overwhelmed [48]. Physicians and patients should play a role in leading AI curriculum development, in addition to data scientists and educators, to ensure the learning outcomes are designed to be clinically useful and patient-centred.

This study had several limitations. The heterogeneity between studies included in this review made it difficult to draw strongly supported conclusions. A lack of follow-up studies evaluating the impacts of current strategies addressing AI education made it difficult to assess their effectiveness. Without these studies, we were also unable to conduct a meta-analysis to strengthen our findings. Lastly, this study focused specifically on AI education in undergraduate medicine, rather than on the continuum of medical education.

Future steps include defining a list of curricular elements that have been validated both by existing literature and domain experts to create a standardized AI curriculum that medical programs can integrate with their current content. We also believe that more cohort studies need to be conducted in order to better evaluate the long-term outcome of AI teachings in undergraduate medicine.

Our study shows that although there is a high degree of heterogeneity among programs and research, the themes of ethics, theory and application, communication, collaboration, quality improvement, and perception and attitudes are recurring. The feedback and attitude of medical trainees towards AI implementation in medical curriculum makes it evident that a systematic approach towards AI education in medicine is warranted.

## Supporting information

**S1 Table. A table displaying the NOS criteria and scoring for all included survey studies.** The NOS criteria is comprised of three categories: selection, comparability, and outcome. (DOCX)

**S2 Table. A table outlining the ROBIS criteria and scoring for all included review studies, with Phase 2 addressing process concerns and Phase 3 assessing bias risk.** (DOCX)

**S1 PRISMA Checklist. PRISMA 2020 Main Checklist.** (PDF)

## Author Contributions

**Conceptualization:** Nikola Pupic, Aryan Ghaffari-zadeh, Ricky Hu, Rohit Singla, Kathryn Darras, Bruce B. Forster.

**Data curation:** Nikola Pupic, Aryan Ghaffari-zadeh, Bruce B. Forster.

**Formal analysis:** Nikola Pupic, Aryan Ghaffari-zadeh, Bruce B. Forster.

**Investigation:** Nikola Pupic, Aryan Ghaffari-zadeh, Rohit Singla, Kathryn Darras, Bruce B. Forster.

**Methodology:** Nikola Pupic, Aryan Ghaffari-zadeh, Ricky Hu, Rohit Singla, Kathryn Darras, Bruce B. Forster.

**Project administration:** Nikola Pupic, Aryan Ghaffari-zadeh, Rohit Singla.

**Resources:** Rohit Singla, Kathryn Darras, Anna Karwowska.

**Supervision:** Rohit Singla, Kathryn Darras, Anna Karwowska, Bruce B. Forster.

**Validation:** Aryan Ghaffari-zadeh, Anna Karwowska.

**Visualization:** Aryan Ghaffari-zadeh, Kathryn Darras.

**Writing – original draft:** Nikola Pupic, Aryan Ghaffari-zadeh.

**Writing – review & editing:** Nikola Pupic, Aryan Ghaffari-zadeh, Anna Karwowska, Bruce B. Forster.

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
