## [Decision Letter · Decision Letter 0]

18 May 2023

PDIG-D-23-00115

Artificial Intelligence Education for Medical Students: A Systematic Review

PLOS Digital Health

Dear Dr. Ghaffari-zadeh,

Thank you for submitting your manuscript to PLOS Digital Health. After careful consideration, we feel that it has merit but does not fully meet PLOS Digital Health's publication criteria as it currently stands. Therefore, we invite you to submit a revised version of the manuscript that addresses the points raised during the review process. Please specifically address the comments from Reviewer 2. 

Please submit your revised manuscript within 60 days Jul 17 2023 11:59PM. If you will need more time than this to complete your revisions, please reply to this message or contact the journal office at digitalhealth@plos.org. Please include the following items when submitting your revised manuscript:

We look forward to receiving your revised manuscript.

Kind regards,

Valentina Lichtner

Academic Editor

PLOS Digital Health

Journal Requirements:

1. Please send a completed 'Competing Interests' statement, including any COIs declared by your co-authors. If you have no competing interests to declare, please state "The authors have declared that no competing interests exist". Otherwise please declare all competing interests beginning with twhe statement "I have read the journal's policy and the authors of this manuscript have the following competing interests:"

2. Please provide separate figure files in .tif or .eps format only and remove any figures embedded in your manuscript file. Please also ensure that all files are under our size limit of 10MB.

3. We ask that a manuscript source file is provided at Revision. Please upload your manuscript file as a .doc, .docx, .rtf or .tex.

4. We have noticed that you have uploaded Supporting Information files, but you have not included a list of legends. Please add a full list of legends for your Supporting Information files after the references list. 

Additional Editor Comments (if provided):

Reviewers' comments:

Reviewer's Responses to Questions

**Comments to the Author**

1. Does this manuscript meet PLOS Digital Health’s publication criteria? Is the manuscript technically sound, and do the data support the conclusions? The manuscript must describe methodologically and ethically rigorous research with conclusions that are appropriately drawn based on the data presented.

Reviewer #1: Yes

Reviewer #2: No

Reviewer #3: Yes

2. Has the statistical analysis been performed appropriately and rigorously?

Reviewer #1: Yes

Reviewer #2: N/A

Reviewer #3: Yes

3. Have the authors made all data underlying the findings in their manuscript fully available (please refer to the Data Availability Statement at the start of the manuscript PDF file)?

Reviewer #1: Yes

Reviewer #2: Yes

Reviewer #3: Yes

4. Is the manuscript presented in an intelligible fashion and written in standard English?

Reviewer #1: Yes

Reviewer #2: Yes

Reviewer #3: Yes

5. Review Comments to the Author

Reviewer #1: This systematic review attempts to identify emerging themes in incorporating artificial intelligence literacy in medical education. The authors do a nice job of synthesizing their findings into relevant, albeit broad, learning objectives for undergraduate medical curricula. 

The study title is too broad and fails to highlight the important goal of this study which was to identify evidence-based recommendations. It is highly recommended to focus the title and align it with the study's objective. 

The methodology is well explained and executed. My only concern is the exclusion of the randomized control trial study design from the review. The rationale and decision to not include RCTs must be clearly explained in the methodology. For example, interventional studies that have looked at the efficacy of utilizing AI for instruction would provide further evidence to contextualize the authors' recommendations and identify more nuanced strengths and weaknesses. 

The discussion is well balanced and adequately states potential challenges in enriching an already saturated curriculum. The article can possibly benefit from a discussion on the role of medical licensing bodies such as the LCME or the Royal College for Physicians and Surgeons of Canada in defining new core competencies on AI or Digital Health Literacy.

Overall, this is an insightful study that identifies key themes to be addressed in an AI-aware medical curriculum.

Reviewer #2: This paper aimed to identify evidence-based recommendations for developing and implementing an AI curriculum in undergraduate medical education through reviewing articles. The authors claimed six themes for the successful implementation of AI in medical school curricula: ethics, theory and application, communication, collaboration, quality improvement, and perception and attitude. Furthermore, they suggested future research about adding formal AI education into medical curricula. 

This paper is very timely and novel. The AI education issue is exciting to global medical educators because of ChagGPT. However, the article needs rigorous efforts for good original content and a valid research methodology.

1) Research gaps 

The authors needed to present research gaps from previous review literature sufficiently. This paper is a systematic review, so the authors should show how many review articles have already existed and which research agenda has been explored in these previous review articles. When the authors present current review aims and methodologies, readers can understand the context of the previous literature and the necessity of this research.

2) Data and analyses 

The data and analyses did not fully support the authors’ claims that the themes are evidence-based. First, the themes were mainly from review articles, but the authors did not evaluate their quality and evidence level. Instead, they only assessed the quality of the cohort and cross-sectional studies based on the Newcastle-Ottawa quality assessment scale. 

Second, the purpose of the previous article did not aim to implement a successful program. Therefore, I don’t think the authors thematized the findings from the included reports. Instead, I guess they thematized the discussion contents from the previous articles. I don't think the discussion contents can be evidence-based.

Therefore, I suggest the authors collect and evaluate articles implementing and evaluating AI programs to support the claims that recommendations are evidence-based. In addition, they should analyze the outcomes of articles related to the research questions.

3) Additional work to improve the paper

Rather than a systematic review, this paper seems more likely to be an integrative or scoping review. The authors wanted to explore implementation strategies using thematic analysis. This aim matches well with the integrative review. 

Or the authors could narrow the focus on survey or review articles to improve the research design.

There are three more things. One is that the results used non-included articles from 50 to 65. I think the authors should remove them. Another is the discussion. AI in healthcare seems unrelated to the research questions in this paper. I hope to see more detailed and profound talks about developing and implementing AI education from the results. The other is Figure 2. As a reader, I would like to see the quality level in Table 1 instead of Figure 2.

4) Research protocol

The authors did not present a registered protocol. So, it would be good for the authors to register the research protocol before starting the study.

5) Guidelines and methodology

This paper needs to describe a more detailed methodology for reproduction. The authors tried to follow the PRISMA guideline, but more was needed. PRISMA 2020 for abstracts and AMSTAR 2 checklist will be helpful. 

First, the abstract should include more detailed information, such as specific inclusion and exclusion criteria, information sources, methods to assess the risk of biases in studies, and limitations of the evidence. 

Second, key research questions did not specify PICO and others, such as time, setting, or study design. 

Third, inclusion criteria only described article types, with no exclusion criteria. I wanted to know whether the authors include non-English articles, experimental studies, or single-institution studies. Besides, I asked whether the authors excluded low-quality articles.

Fourth, two or more researchers should select articles independently. The authors described two researchers conducting study screening, data extraction, and thematic analysis. However, they should have commented on how many researchers perform study selection and quality assessment.

Fifth, I wondered why the authors chose NOS because NOS can evaluate only cohort and case-control studies. Many other evaluation tools exist, such as AMSTAR, MERSQI, or BEME. This paper included various articles, so I suggest the authors use more appropriate tools for surveys and review articles.

6) Organization of the manuscript

The manuscript was understandable, but needed to check for omissions and errors. In addition, for readers who are not familiar with article evaluation tools, NOS only is difficult to understand. 

Furthermore, the authors said that they separated the previous studies into survey and interventional. Interventional studies were mainly reports and reviews, which confused me because interventional studies usually mean experimental studies. Therefore, I suggest the authors change the term ‘interventional’ and explain the definition of it.

Reviewer #3: Pupic et al. did a systematic review on the AI-specific education in medical schools. They had collected more than 900 original paper and scoped down into 30 ones fit the study criteria. In conclusion they found six core themes should be suggested implant into the future AI related medical education. 

Minor points:

1. It is also mentioned some field of medical education developed more dependent on current state of art technique of AI, while other filed less developed. So, it is suggested to draw a bar chart to demonstrate the diversity of the current AI application situation in various medial field or specialty. And the distribution of the 30 selected paper should also be more elucidated on each research field. 

2. The specific indicators of quality assessment and statistical analysis should be clearer. The current presentation format of Table 1 and Figure 2 and 3 are not strait forward and difficult to link with the result part. 

In all, the topic is meaningful and the research method was thoroughly with high efficiency and plentiful of data. The current manuscript should be accepted after the minor revisions.

6. PLOS authors have the option to publish the peer review history of their article (what does this mean?). If published, this will include your full peer review and any attached files.

**Do you want your identity to be public for this peer review?** For information about this choice, including consent withdrawal, please see our Privacy Policy.

Reviewer #1: No

Reviewer #2: Yes: HyeRin Roh

Reviewer #3: Yes: Dawei Yang

---

## [Editor Report · Decision Letter 1]

14 Sep 2023

An Evidence-Based Approach to Artificial Intelligence Education for Medical Students: A Systematic Review

PDIG-D-23-00115R1

Dear Mr Ghaffari-zadeh,

We are pleased to inform you that your manuscript 'An Evidence-Based Approach to Artificial Intelligence Education for Medical Students: A Systematic Review' has been provisionally accepted for publication in PLOS Digital Health.

Best regards,

Valentina Lichtner

Academic Editor

PLOS Digital Health